



# Role of chemical production and depositional losses on formaldehyde in the Community Regional Atmospheric Chemistry Multiphase Mechanism (CRACMM)

T. Nash Skipper[1,2], Emma L. D'Ambro[2], Forwood C. Wiser[3], V. Faye McNeill[3,4], Rebecca H. Schwantes[5],
Barron H. Henderson[6], Ivan R. Piletic[2], Colleen B. Baublitz[6], Jesse O. Bash[2], Andrew R. Whitehill[2],
Lukas C. Valin[2], Asher P. Mouat[7], Jennifer Kaiser[7,8], Glenn M. Wolfe[9], Jason M. St. Clair[9,10], Thomas F.
Hanisco[9], Alan Fried[11], Bryan K. Place[9,12], and Havala O.T. Pye[2]

[1] Oak Ridge Institute for Science and Education, Office of Research and Development, U.S. Environmental Protection Agency, Research Triangle Park, North Carolina, USA
[2] Office of Research and Development, U.S. Environmental Protection Agency, Research Triangle Park, North Carolina, USA
[3] Department of Chemical Engineering, Columbia University, New York, New York, USA
[4] Department of Earth and Environmental Sciences, Columbia University, New York, New York, USA
[5] Chemical Sciences Laboratory, National Oceanic and Atmospheric Administration, Boulder, Colorado, USA
[6] Office of Air and Radiation, U.S. Environmental Protection Agency, Research Triangle Park, North Carolina, USA
[7] School of Civil and Environmental Engineering, Georgia Institute of Technology, Atlanta, GA, USA
[8] School of Earth and Atmospheric Sciences, Georgia Institute of Technology, Atlanta, GA, USA
[9] Atmospheric Chemistry and Dynamics Laboratory, NASA Goddard Space Flight Center, Greenbelt, MD, USA
[10] Joint Center for Earth Systems Technology, University of Maryland Baltimore County, Baltimore, MD, USA
[11] Institute of Arctic and Alpine Research (INSTAAR), University of Colorado, Boulder, CO, USA
[12] SciGlob Instruments and Services, LLC, Columbia, MD, USA

*Correspondence to*: Havala O.T. Pye (pye.havala@epa.gov)

**Abstract.** Formaldehyde (HCHO) is an important air pollutant due to its direct health effects as an air toxic that contributes to elevated cancer risk, its role in ozone formation, and its role as a product from oxidation of most gas phase reactive organic carbon. We make several updates affecting secondary production of HCHO in the Community Regional Atmospheric Chemistry Multiphase Mechanism (CRACMM) in the Community Multiscale Air Quality (CMAQ) model. Secondary HCHO from isoprene and monoterpenes is increased, correcting an underestimate in the current version. Simulated 2019 June–August surface HCHO during peak photochemical production (11am–3pm) increased by 0.6 ppb (32%) over the southeastern US and by 0.2 ppb (13%) over the entire contiguous US. The increased HCHO compares more favorably with satellite-based observations from TROPOMI and observations from an aircraft campaign. Evaluation against hourly surface observations indicates a missing nighttime sink for HCHO which can be ameliorated by adding bidirectional exchange of HCHO and a leaf wetness dependent deposition process which increases nighttime deposition, decreasing 2019 June–August nocturnal (8pm–4am) surface HCHO by 1.1 ppb (36%) over the southeastern US and 0.5 ppb (29%) over the entire contiguous US. The ability of CRACMM to capture peak levels of HCHO at midday is improved, particularly at sites in the northeastern US, while peak levels at southeastern US sites are improved though still lower than observed. Using established risk assessment methods, lifetime exposure of the contiguous U.S. population (~320 million) to ambient HCHO levels predicted here may result in 6200



lifetime cancer cases, 40% of which are from controllable anthropogenic emissions of nitrogen oxides and reactive organic compounds. Chemistry updates will be available in CRACMM version 2 (CRACMM2) in CMAQv5.5.

## 1 Introduction

Formaldehyde (HCHO) is a gas-phase reactive organic compound designated as a hazardous air pollutant (HAP) by the U.S.
Environmental Protection Agency (EPA). It is among the top three species contributing to noncancer health risk and the leading driver of cancer risk from ambient exposure to inhaled air toxics in the United States (Scheffe et al., 2016; Strum and Scheffe, 2016). EPA's 2019 AirToxScreen assessment estimates a nationwide average cancer risk of ~15 in a million for HCHO, about half of the total national average cancer risk from ambient exposure to air toxics (https://www.epa.gov/AirToxScreen). HCHO is also an important oxidation product and indicator of gas-phase chemistry. Once formed, HCHO can be a source of radicals
that modulate cycling of nitrogen oxides ($NO_X=NO+NO_2$) and thus formation of the criteria pollutant ozone ($O_3$). HCHO is quantified through remote sensing and has been used to provide top-down constraints on emissions of isoprene and other precursor species through inverse modeling (Fortems-Cheiney et al., 2012; Kaiser et al., 2018; Oomen et al., 2024) and, along with satellite-based observations of $NO_2$, to characterize $O_3$ chemical regimes (Martin et al., 2004; Duncan et al., 2010; Tao et al., 2022).


The abundance of ambient HCHO is influenced by both primary emissions of HCHO and its precursors as well as atmospheric chemistry. Primary HCHO is emitted by many sources as a combustion byproduct as well as from natural sources. Biogenic sources such as vegetation are the largest source of primary HCHO in the US (~1200 Gg yr$^{-1}$) with other major sources including fires (~300 Gg yr$^{-1}$), mobile sources (~40 Gg yr$^{-1}$), and wood burning for residential heating (~20 Gg yr$^{-1}$) (Foley et
al. (2023) based on 2017 National Emissions Inventory (NEI), Fig. S1). HCHO's short lifetime of only a few hours against photolysis and reaction with the OH radical means impacts of primary HCHO are typically localized near source (characteristic transport of ~30 km for 3 h lifetime with 3 m s$^{-1}$ wind speed). Secondary production tends to dominate over primary sources in driving total abundance, particularly in warmer months when HCHO levels are the highest (Dix et al., 2023). HCHO is produced from oxidation of nearly every gas-phase reactive organic carbon (ROC) species with isoprene being the biggest
source of secondary HCHO. Other important precursors include methane and alkenes (Luecken et al., 2012).

EPA's AirToxScreen as well as inverse modeling for emission estimation rely on chemical transport models (CTMs) to simulate HCHO. Specifically, as part of AirToxScreen, ambient exposure levels of air toxics are obtained from concentrations predicted by the Community Multiscale Air Quality (CMAQ) CTM combined with local scale information from a dispersion
model (U.S. EPA, 2022b), and CMAQ alone provides the estimates of secondary HCHO. CMAQ has been previously reported to underestimate HCHO (Luecken et al., 2012; Luecken et al., 2018) which could propagate to errors in predictions of health risk. Combined with the national population for 2019, the AirToxScreen nationwide cancer risk from HCHO (2019 value)




implies ~4800 cancer cases result from lifetime exposure. However, Zhu et al. (2017) estimated between 6600 and 12500 cancer cases based on exposures derived from satellite-based HCHO observations. A more accurate representation of secondary HCHO could improve inverse modeling estimates of emissions as well as our understanding of the role of ambient HCHO in inhalation health risks.

Here, we focus on the representation of secondary HCHO production in the Community Regional Atmospheric Chemistry Multiphase Mechanism (CRACMM). CRACMM is designed to integrate modeling of $O_3$, $PM_{2.5}$, and HAPs and has been primarily applied in CMAQ (Pye et al., 2023). We make several updates to CRACMM version 1 (CRACMM1), leading to CRACMM version 2 (CRACMM2). Most of the updates in CRACMM2 target HCHO, and additional updates for completeness are documented here for users of CMAQ and CRACMM2. Chemistry updates were screened with a box model, the Framework for 0-D Atmospheric Modeling (F0AM) (Wolfe et al., 2016a), and then tested in a series of regional CMAQ simulations covering the contiguous US (CONUS). The performance of CRACMM (1 and 2) in CMAQ are evaluated with a suite of observations including satellite based HCHO from TROPOspheric Monitoring Instrument (TROPOMI), observations from an aircraft campaign, and hourly surface observations from several field deployments. Based on the evaluation, sensitivity simulations are conducted to explore areas for future improvement of HCHO in CMAQ CRACMM. Estimates of cancer risk from ambient exposure to HCHO derived from CMAQ CRACMM are provided along with an estimate of the portion of cancer risk that is controllable through reductions in anthropogenic $NO_X$ and ROC emissions.

## 2 Box model simulations

Box model simulations were conducted using F0AM (Wolfe et al., 2016a) v4.3 to explore the representation of secondary production of HCHO in CRACMM1 compared to the Master Chemical Mechanism (MCM) v3.3.1 (Jenkin et al., 1997; Saunders et al., 2003; Jenkin et al., 2003; Bloss et al., 2005; Jenkin et al., 2012; Jenkin et al., 2015). Results from MCM are used as a benchmark to compare with CRACMM1 since it provides a much more detailed representation of chemistry (17224 reactions and 5832 species in MCM compared to 508 reactions and 229 species in CRACMM1). The box model simulations serve as a screening level identification of precursor systems that may not produce sufficient secondary HCHO in CRACMM1. Discrepancies between MCM and CRACMM1 indicate differences in mechanism assumptions but not necessarily an error in CRACMM1. Emission sectors and/or precursors systems that showed meaningful difference from MCM were used to prioritize chemical systems for further analysis and development in CRACMM2. F0AM was run as a batch simulation with pressure, relative humidity (RH), and temperature held at 1013 mbar, 10%, and 298 K, respectively. Photolysis rates from CRACMM1 were matched to existing MCM photolysis rates in F0AM. Simulations were run for 8 hours of photochemical processing with $NO_X$ initialized at 1 ppb of $NO_2$ and allowed to evolve freely during the simulation. Effects of OH-initiated oxidation and ozonolysis were tested separately (oxidant concentrations held constant at $10^6$ molecules cm$^{-3}$ OH and zero $O_3$ in Fig. 1 or 30 ppb $O_3$ and zero OH in Fig. S3). Simulations including both OH and $O_3$ were also conducted (Fig. S4).




Initial concentrations of ROC precursors were set based on grouping emissions in two different ways: by emissions sector and by precursor system. Emissions from each anthropogenic emissions sector for individual species available in the EPA SPECIATE database (Simon et al., 2010) as previously compiled by Pye et al. (2023) were mapped to species available in CRACMM and MCM. The concentrations of ROC precursors were initialized based on the emissions of each species with

100 Gg of annual emissions represented by 1 ppb (except for primary HCHO which was excluded). For the emission sector simulations (Fig. 1a), all emitted ROC species from each of 20 emissions sectors (Table S1) were initialized at their emission-weighted value. For the precursor system simulations (Fig. 1b), the total emissions across all sectors were divided into 19 distinct precursor groups (Table S2), and a simulation was conducted with initial concentrations for only the species belonging to a particular precursor group. Secondary HCHO from biogenic emissions was similarly assessed except that initial precursor

concentrations were set with 1000 Gg of annual emissions represented as 1 ppb. At the end of 8 hours of photochemical processing, the ending HCHO concentrations simulated by MCM and CRACMM1 were used to compare the representation of secondary HCHO from CRACMM1 and MCM. We also provide here for comparison the secondary HCHO simulated in CRACMM2 after all chemistry updates (Sect. 3) were added. These ending concentrations of HCHO are not intended to represent the expected contribution to ambient HCHO from a particular emissions sector or precursor group; they are only

intended to serve as a convenient metric to compare secondary production of HCHO across mechanisms and to identify systems requiring further investigation. The emissions-weighted approach used here for setting the initial ROC precursor concentrations means that the magnitude of the ending concentration of HCHO depends on two major factors: the total ROC emissions from the individual emission sector or precursor system (Fig. S2) and the yield of HCHO from the ROC species included in the simulations. The box model setup employed here is limited in its ability to assess some atmospheric processes,

such as transport or interactions between emissions from different sectors. However, it offers an efficient way to conduct idealized tests of HCHO production with different chemical mechanisms.

Results from the F0AM box model simulations with OH oxidation are summarized in Fig. 1. Secondary HCHO from biogenic sources is much higher in MCM compared to CRACMM1. This discrepancy is mostly from isoprene which has much lower

HCHO production in CRACMM1, though monoterpenes also contribute. Secondary formation of HCHO from isoprene in CRACMM1 is inherited from RACM2, which has been found to produce less HCHO from isoprene compared to other mechanisms (Wolfe et al., 2016a; Wiser et al., 2023). Production from sesquiterpenes is also underestimated, but this is less influential since sesquiterpene emissions are small relative to isoprene and monoterpenes. Isoprene updates in CRACMM2, specifically an increased HCHO yield from isoprene oxidation, drive substantially higher (~a factor of 6) secondary HCHO

from total biogenic emissions compared to CRACMM1 which is more consistent with MCM. HCHO from monoterpenes is also increased in CRACMM2 and is more in line with what is predicted by MCM. The production of HCHO from isoprene with the AMOREv1.2 condensed mechanism has also been compared with the detailed isoprene mechanism from Wennberg et al. (2018) (see SI for more details).





Other than biogenic emissions, fires have the highest secondary HCHO production by sector in these tests because they have large total ROC emissions. Secondary HCHO simulated by MCM for fire sectors is higher than CRACMM1, primarily due to differences in secondary HCHO from alkenes mostly in the form of terminal olefins. HCHO from volatile chemical products (VCPs) was identified as an important source of difference between mechanisms where HCHO from CRACMM1 was low compared to MCM. The largest source of secondary HCHO for VCPs was from limonene. Updates to the limonene system

(Sect. 3.4) resulted in better agreement between CRACMM2 and MCM-estimated secondary HCHO. Gasoline mobile sources (onroad gas and nonroad gas) and the nonpt sector (a miscellaneous sector for area sources that do not have their own sector) were also low in CRACMM compared to MCM, mostly due to alkenes. HCHO from non-EGU point sources (also sometimes called the ptnonipm sector) was also underestimated in the box model testing. Part of the underestimate for the non-EGU point sector was from the representation of styrene which was lumped with CRACMM1 species XYM (represented with the

chemistry of m-xylene) but added as a new explicit species in CRACMM2 (Sect. 3.5). Styrene made up 65% of emissions mapped to XYM for the ptnonipm sector, which was a much larger fraction than for other sectors (e.g., 12% for fires, 6% for VCPs, and 1% for gasoline-powered mobile sources). The addition of explicit styrene improved the comparison between MCM and CRACMM2 because the HCHO yield from styrene is much greater than that of m-xylene; however, secondary HCHO from this sector is still low compared to MCM. The other sectors mostly had good agreement between MCM and CRACMM

with most of the secondary HCHO production driven by alkenes. One exception is the agricultural sector (ag) where dimethyl sulfide (DMS) contributed to HCHO for MCM but is not currently represented in CRACMM.

When total emissions across all sectors (excluding biogenic emissions) are separated into compound precursor groups, alkenes, such as ethene and propene, make up the largest contribution to secondary HCHO. Ethene is represented explicitly in

CRACMM and has very similar HCHO production in MCM and CRACMM. However, the lumped terminal alkene species (OLT) in CRACMM has a lower HCHO yield (0.78) than the effective HCHO yield of propene in MCM (0.98), leading to lower secondary HCHO from alkenes in CRACMM1 compared to MCM which contributes to the low secondary HCHO seen in several source sectors. Alkene chemistry for terminal and internal olefins has not been modified in CRACMM1 or 2 since the original RACM2 implementation but is an area where future development may be needed.


Secondary HCHO from monoterpenes (which are represented in MCM by α-pinene, β-pinene, and limonene) is low in CRACMM1 compared to MCM and has been improved with CRACMM2. Some other groups with lower HCHO in CRACMM include furans, alcohols, and ketones. These are responsible for a smaller fraction of total ROC emissions and were not prioritized for updates in CRACMM2. Aldehydes stand out as a group where secondary HCHO in CRACMM was higher than

in MCM which was a result of higher production in CRACMM of methylperoxy radicals, which produces HCHO through reaction with NO. The "other ROC" group is dominated by semi-volatile and intermediate volatility compounds (generally





C12 and larger species) which are important for SOA formation in CRACMM but do not exist in MCM and thus do not produce HCHO in MCM.




**Figure 1. Ending HCHO concentration after 8-h box model simulations for MCM, CRACMM1, and CRACMM2 separated by emissions sector (a) and ROC precursor group (b). Results shown here are for a simulation where OH was held constant at $10^6$ molecules cm$^{-3}$ while O$_3$ was held at zero. Each bar represents a separate box model simulation with initial ROC precursor concentrations dependent on the emissions sector or precursor group. Descriptions of the emission sectors and of the species included in each precursor group are given in Tables S1-S2.**

## 3 Chemistry updates

CRACMM2 includes updates to several chemical systems which are discussed below. CRACMM1 is described in detail by Pye et al. (2023) and has been applied within CMAQ for the northeastern US to investigate O$_3$ (Place et al., 2023), CONUS during summer to investigate PM$_{2.5}$ (Vannucci et al., 2024), and CONUS to investigate SOA from asphalt paving (Seltzer et





al., 2023). Some relevant details on CRACMM1 chemistry are given here for comparison with CRACMM2. A list of all reactions that have been updated or added in CRACMM2 are provided in Table S3. In total, the number of reactions is increased from 508 to 531 and the number of species (gases and particles) is increased from 229 to 239 in CRACMM2 compared to CRACMM1.

## 3.1 AMORE isoprene

CRACMM1 included two options for isoprene chemistry. The main mechanism used isoprene chemistry based on RACM2 (Goliff et al., 2013; Sarwar et al., 2013) with additional IEPOX chemistry which is not included in the native RACM2 representation. A separate version of CRACMM included the Automated Model Reduction (AMORE) version 1.0 (Wiser et al., 2023) condensation of a detailed isoprene mechanism (Wennberg et al., 2018) and was referred to as CRACMM1AMORE in CMAQ. The development of the AMORE mechanism condensation technique is described in detail by Wiser et al. (2023). Briefly, AMORE takes the full mechanism along with a list of priority species, estimates the sensitivity of the full mechanism to variations in important species, and builds a reduced mechanism which emulates the sensitivity of the full mechanism. The AMOREv1.1 representation of isoprene chemistry was implemented in GEOS-Chem which yielded similar model performance with improved computational speed compared to the default GEOS-Chem mechanism (Yang et al., 2023). AMOREv1.2 is the default isoprene representation in CRACMM2, merging the base CRACMM and CRACMM-AMORE mechanisms, to better represent isoprene degradation productions and SOA precursors. AMOREv1.2, developed here (see SI for additional details), is intended to produce better $NO_X$ cycling and $O_3$ compared to CRACMM1AMORE and adds new SOA precursors. New gas phase species are INO2 (isoprene nitrooxy peroxy radical), IPX (lumped isoprene tetrafunctional compounds), and INALD (lumped isoprene nitrates). Two new SOA species were added as products of heterogeneous uptake of IPX and INALD (see Sect. 3.3 for details). In addition, HCHO yields were updated to more closely follow the detailed mechanism by Wennberg et al. (2018) based on box model testing (Fig. S5).

## 3.2 Methane

CMAQ specifies a fixed global background methane value of 1850 ppb by default, although the level can be modified by the user. CRACMM1 includes one methane reaction with OH, but the Carbon Bond family of mechanisms in CMAQ also include a reactive tracer species for emitted methane (ECH4) to capture the effects of local methane emissions on top of the global background. CRACMM2 adds the ECH4 species and includes a single ECH4 reaction with OH that is identical to the global methane reaction with OH from CRACMM1. Adding ECH4 can have small effects on secondary HCHO production as the methyl peroxy radical (MO2) produced from ECH4 + OH is a source of HCHO through reaction with NO and peroxy radical ($RO_2$) cross reactions. ECH4 is typically small compared to the global background methane value of 1850 ppb and only has notable impacts on other model species near sources with high ECH4 emissions.





## 3.3 Heterogeneous uptake

Four new heterogeneous uptake pathways have been added to CRACMM2. Two of these are heterogeneous uptake of isoprene-derived species from AMORE to form SOA. Lumped isoprene tetrafunctional compounds (IPX) form AISO4, and lumped isoprene nitrates (INALD) form AISO5 and nitric acid. Uptake of these species is expected to behave similarly to IEPOX

uptake, so we base their uptake rates on the existing IEPOX uptake rate in CMAQ (Pye et al., 2013; Pye et al., 2017). Uptake of IPX is scaled to two times the IEPOX uptake rate, and uptake of INALD is scaled to half of the IEPOX uptake rate. We also add heterogeneous uptake of $HO_2$ and nitrate radicals in CRACMM2. Heterogeneous uptake of $HO_2$ radicals has been included in other CTMs but not yet in any released version of CMAQ. Uptake of $HO_2$ tends to reduce $O_3$ and may be particularly important when aerosol concentrations are very high (Ivatt et al., 2022). CRACMM2 uses an uptake coefficient of

$\gamma=0.2$ and assumes that $HO_2$ produces only water (Ivatt et al., 2022). More complex parameterizations involving $HO_2$ uptake catalyzed by copper and iron have sometimes been employed (Mao et al., 2013), but the simpler version that we have opted for in CRACMM2 is commonly employed in other CTMs such as GEOS-Chem. Heterogeneous uptake is a potential sink for nitrate radicals which may influence nighttime chemistry when oxidation by nitrate radicals takes place. CRACMM2 uses an uptake coefficient of $\gamma=10^{-3}$ for nitrate and assumes that nitrate radical uptake produces nitric acid (Jacob, 2000; Zhu et al.,

2024). These heterogeneous uptake additions are not targeted towards improving HCHO but are implemented primarily for their effects on SOA (uptake of isoprene-derived compounds) and $O_3$ (radical uptake).

## 3.4 Monoterpenes

Monoterpenes in CRACMM are categorized based on their number of double bonds as either API (represented with the chemistry of α-pinene) or LIM (represented with the chemistry of limonene). Monoterpene chemistry in CRACMM1 was

largely based on MCM with additional updates including autoxidation pathways. After α-pinene, β-pinene is one of the most abundant monoterpenes from biogenic emissions (Guenther et al., 2012). In CRACMM, both α-pinene and β-pinene are represented by the lumped species API; however, the yield of HCHO from these monoterpenes differs significantly as the presence of the exocyclic terminal double bond in β-pinene leads to greater HCHO production. Experimental yields of HCHO from α-pinene have been reported as 0.16–0.23 (Nozière et al., 1999; Orlando et al., 2000; Lee et al., 2006) while yields from

β-pinene have been reported as 0.45–0.53 (Hatakeyama et al., 1991; Orlando et al., 2000; Lee et al., 2006). HCHO from API in CRACMM1 is underestimated in part because the larger yield from β-pinene is not accounted for. Limonene HCHO yields are also likely underestimated as the yield in CRACMM1 (0.28) is less than what has been reported in experimental results (0.43) (Lee et al., 2006).

In CRACMM2, monoterpene chemistry has been updated based on work by Schwantes et al. (2020) which primarily used experimental results to determine monoterpene oxidation products and yields as part of the development of an update to the Model of Ozone And Related chemical Tracers (MOZART) chemical mechanism (Emmons et al., 2020). In the updated





MOZART mechanism (MOZART-TS2), monoterpenes are grouped into four species represented by α-pinene, β-pinene, limonene, and myrcene which each have unique detailed chemical evolution. Some deviations and simplifications are made in

porting the MOZART-TS2 chemistry to CRACMM2. To manage the computational burden of CRACMM2, we retain the two monoterpene species from CRACMM1 (API and LIM) and map species from MOZART-TS2 to existing species from CRACMM1. We retain the behavior from CRACMM1 such that a fraction of the peroxy radicals formed from oxidation of a monoterpene by OH or nitrate (2.5% for API; 5.5% for LIM) undergo rapid autoxidation based on Piletic and Kleindienst (2022). The products from the remaining monoterpene peroxy radicals (i.e., those that do not undergo rapid autoxidation) and

from ozonolysis of monoterpenes are updated based on Schwantes et al. (2020).

API products are updated to include products from both α-pinene and β-pinene. We assume 65% of products are from α-pinene and 35% are from β-pinene based on the relative emissions of these species estimated by the Biogenic Emission Inventory System (BEIS) over the 12 km CONUS modeling domain (Fig. S8). We avoid adding a new β-pinene species to CRACMM2

because it requires adding around 30 new reactions to represent β-pinene oxidation and $RO_2$ fate which was deemed too computationally expensive and because the reactivity of α-pinene and β-pinene are similar enough to be represented with the same reaction for both species. The updates to monoterpene reactions and products are detailed in Table S3. Most notably for HCHO, the effective yield from API+OH $RO_2$ (APIP1) has increased from 0 to 0.31 for $RO_2$+NO; from 0 to 0.40 for $RO_2$+NO$_3$, and from 0 to 0.06 for $RO_2$+HO$_2$. HCHO yields for $RO_2$+$RO_2$ cross reactions involving APIP1 have also increased. HCHO

yields increased from 0 to 0.46 for API ozonolysis. In CRACMM1, HCHO from API was exclusively due to later generation chemistry involving pinonaldehyde (species PINAL). CRACMM2 forms HCHO in earlier generations and brings HCHO yields more in line with experimental yields. LIM products are updated based on the limonene representation from MOZART-TS2 where the most significant updates for HCHO are an increase in the yield of HCHO from LIM+OH $RO_2$ (LIMP1) from 0.28 to 0.43 for $RO_2$+NO and an increase from 0 to 0.33 for LIM ozonolysis.


CRACMM1 includes two monoterpene aldehydes based on pinonaldehyde (species PINAL) and limonaldehyde (species LIMAL) which react with OH to produce peroxy radicals (PINALP and LIMALP) and acyl peroxy radicals (species RCO3). PINALP and LIMALP react with NO and HO$_2$ but can also form highly oxygenated organic molecules (HOM) with an autoxidation rate of 1 s$^{-1}$ in CRACMM1. Box model testing indicated that this autoxidation rate made the bimolecular NO and

HO$_2$ channels uncompetitive at typical atmospheric levels of NO and HO$_2$ (i.e., essentially all PINALP and LIMALP would autoxidize and make HOM), so the autoxidation rates are updated for CRACMM2. The autoxidation of PINAL and LIMAL will proceed via multiple steps involving slightly different mechanistic pathways because of differences in chemical structure. All H-shift rates are approximated using the structure activity relationships developed by Vereecken and Nozière (2020). Specifically, an OH initiation reaction with PINAL will produce an acyl peroxide radical as the dominant product (represented

by CRACMM species PINALP) while the same reaction will produce a tertiary peroxy radical via OH addition to the double bond in LIMAL. For LIMAL, a subsequent 1,6-H shift that abstracts the aldehyde H at a rate of 0.29 s$^{-1}$ gives rise to an



analogous albeit more oxidized acyl peroxide radical (represented by CRACMM species LIMALP). At this point, both acyl peroxy radicals will likely abstract from a tertiary carbon via a 1,5-H shift that is fairly rapid (0.7 s$^{-1}$) and a subsequent 1,5-H shift from the β-oxo site produces HOM radicals at a rate of 0.02 – 0.03 s$^{-1}$. Given that the latest generation autoxidation

reaction is the slowest, it was used to approximate the overall autoxidation rate. This approximation simplifies the modeled autoxidation process because alternative pathways may exist including cyclobutyl ring opening following H abstraction for PINAL (Iyer et al., 2021) or peroxy radical ring closure reactions for LIMAL if the initiation step extracts the aldehyde H (Piletic and Kleindienst, 2022). Within this approximation, the autoxidation rates of monoterpene aldehydes have been updated in CRACMM2 to 0.029 s$^{-1}$ for PINALP and 0.024 s$^{-1}$ for LIMALP. At these autoxidation rates, reaction with NO or HO$_2$

becomes competitive with HOM formation. The rates and products of PINALP and LIMALP reactions with NO and HO$_2$ have also been updated based on parameterizations from Wennberg et al. (2018). For the monoterpene systems, autoxidation occurs in both the first and second (through aldehydes) generation chemistry. Since autoxidation is an efficient source of SOA in monoterpene systems, balancing the role of autoxidation across generations is needed to ensure reasonable SOA production. API ozonolysis in CRACMM2 retains a prompt (first generation) autoxidation channel with a yield of 0.21 for an RO$_2$ with

an autoxidation rate set to the PINALP rate rather than a fixed yield (no competition with biomolecular RO$_2$ reactions) of 5% for HOM-RO$_2$ as in CRACMM1. Aldehyde yields are significantly higher in the LIM ozonolysis system compared to API, and all autoxidation from LIM ozonolysis was tied to further aldehyde reaction. Future work should aim to improve the representation of autoxidation across monoterpene ozonolysis and aldehyde systems.

CRACMM1 contains one monoterpene nitrate species (TRPN) which forms primarily from reactions of API and LIM derived peroxy radicals with NO. Further oxidation of TRPN in CRACMM1 results in a 100% yield of HOM, though deposition of TRPN is a competing fate which reduces the effective SOA yield from TRPN in CTMs. In CRACMM2, several additional chemical fates are added for TRPN. Photolysis of TRPN is added, resulting in recycling of NO$_X$ and smaller organic products (species KET and UALD). Oxidation of TRPN no longer produces HOM; instead, we assume that oxidation of TRPN results

in a 33% yield of a second-generation monoterpene nitrate species (new species HONIT). The remaining 67% of products result in the release of the nitrate group to NO$_2$ plus fragmentation products. The 1/3 to 2/3 split to HONIT assumes that 1/3 of TRPN is unsaturated (i.e., contains a double bond) while the remaining two thirds are saturated following a monoterpene nitrate mechanism previously implemented in GEOS-Chem by Fisher et al. (2016) based on a mechanism by Browne et al. (2014). Unsaturated monoterpene nitrates are expected to retain the nitrate group and form a more oxygenated monoterpene

nitrate upon reaction while saturated monoterpene nitrates are expected to release the nitrate group to form NO$_2$ plus other fragmentation products. Limonene oxidation by OH is expected to produce only unsaturated products based on Fisher et al. (2016), so no fragmentation products from limonene derived nitrates are expected. Thus, fragmentation products are based on oxidation products of α-pinene and β-pinene derived nitrates in MCM. The α-pinene nitrate products from MCM indicate a 62% yield of pinonaldehyde (species PINAL) and 38% yield of a ketone (species KET). The β-pinene nitrate products from

MCM indicate a 92% yield of a ketone (species KET) and HCHO and 8% yield of an aldehyde (species ALD). Since α-pinene



and β-pinene are lumped in species API, we apply a 65/35 split of α-pinene and β-pinene based on the biogenic emissions of these species to calculate the total yields of these additional products.

The new second generation monoterpene species HONIT can be lost through photolysis, reaction with OH, deposition, or hydrolysis. Both TRPN and HONIT are treated as semivolatile species in CRACMM2 with C* of ~1400 µg m$^{-3}$ and ~0.04 µg m$^{-3}$ respectively based on their structures (Pankow and Asher, 2008). The resulting SOA from TRPN and HONIT are tracked as two new aerosol species (ATRPN and AHONIT). These monoterpene nitrate aerosol species also undergo hydrolysis with 3-h lifetime to form aerosol HOM (species AHOM) and nitric acid (Pye et al., 2015). The updates to monoterpene nitrates do not have significant effects on HCHO, but the updates to other parts of the monoterpene system offered an opportunity to address these additional areas that were known to be missing from CRACMM1.

### 3.5 Aromatics

The chemistry of aromatics in CRACMM1 is generally based on MCM and the work of Xu et al. (2020) as described in Pye et al. (2023). In CRACMM2 most aromatic species are unchanged from CRACMM1, but we make some updates to how emissions of aromatic compounds are mapped to lumped mechanism species. CRACMM1 includes two lumped xylene-based species defined by a range in OH reactivity: XYE includes ethylbenzene, o- and p-xylene, and other aromatic species with chemistry based on ethylbenzene and XYM includes m-xylene and other aromatic species with chemistry based on m-xylene. In CRACMM1, single ring aromatic species benzene, toluene, and those in the intermediate-volatility range are separately represented from XYE and XYM. In CRACMM2, XYE is renamed to EBZ to represent ethylbenzene explicitly and no longer includes any xylene isomers or other species. XYM is renamed to XYL and is now used to represent all isomers of xylene plus other single ring aromatic species that are not otherwise represented. Isomers of xylene are commonly reported in measurements as a mixture of o-, m-, and/or p-xylene. Lumping all xylenes into XYL prevents artificial distinctions in reactivity that are not actually available from measurements or the emission inventories informed by them. There are no changes in the chemistry of XYE (now EBZ) or XYM (now XYL) between CRACMM1 and CRACMM2 – only changes in how emissions are mapped onto these species.

Styrene is lumped into XYM in CRACMM1 but has been separated as an explicit species STY in CRACMM2. Styrene was added because it is a HAP and because it also has a much higher yield of secondary HCHO than m-xylene which led to underestimates in secondary HCHO estimated by box modeling (Sect. 2). Styrene chemistry is based exclusively on MCM and proceeds through one route: OH addition to the exocyclic double bond (Jenkin et al., 2003; Bloss et al., 2005). Molteni et al. (2018) quantified HOM yields from aromatics but did not include styrene in their tests. Since autoxidation in aromatic systems likely occurs for bicyclic RO$_2$ (Molteni et al., 2018; Xu et al., 2020) which does not occur in the styrene system, we assume first generation styrene products do not undergo autoxidation. No organic nitrates are predicted. Major products include HCHO, benzaldehyde, and peroxides. The peroxide is predicted to have a C* of $1.5 \times 10^3$ µg m$^{-3}$ according to EPISuite vapor



pressure so it is mapped to an oxygenated IVOC with O:C=0.2 (species VROCP3OXY2). VROCP3OXY2 undergoes
345   multigenerational oxidation leading to fragmentation products as well as SOA. Previous work (Tajuelo et al., 2019; Yu et al.,
2022) suggests styrene produces SOA in small amounts which are not considered here, although VROCP3OXY2 can go on to
make SOA in further generation chemistry.

## 4 CMAQ simulations

CTM simulations were conducted using CMAQv5.4 (U.S. EPA, 2022a) and model inputs from the EQUATES (EPA's Air
350   Quality TimE Series) modeling framework (Foley et al., 2023). The CMAQ model setup is the same as described in Vannucci
et al. (2024). The modeling domain covers the CONUS with a horizontal resolution of 12 km. Meteorological inputs are from
the Weather Research Forecasting (WRF) model version 4.1.1 (Skamarock et al., 2019) processed through the Meteorology-
Chemistry Interface Processor (MCIP) (Otte and Pleim, 2010) for use in CMAQ. Boundary and initial conditions were from a
2019 northern hemispheric simulation from EQUATES with species from the Carbon Bond 6 mechanism mapped to
corresponding CRACMM species. Emissions from EQUATES were processed through SMOKE to generate model-ready
emission inputs with CRACMM emission speciation. Mapping of emissions species to model species uses the Detailed
Emissions Scaling, Isolation, and Diagnostic (DESID) module in CMAQ (Murphy et al., 2021). The emissions mapping step
is particularly important in CRACMM for applying appropriate volatility profiles to emissions of primary organic carbon and
non-carbon organic matter as operational inventories currently lack that information. Biogenic emissions are computed inline
in CMAQ using the BEIS module (Bash et al., 2016). The Surface Tiled Aerosol and Gaseous Exchange (STAGE) dry
deposition model is used (Appel et al., 2021; Clifton et al., 2023). Annual simulations for 2019 were conducted using the base
CRACMM1 mechanism and the updated CRACMM2 mechanism with one month spin up in December 2018 to reduce the
influence of initial conditions. The incremental impacts of chemistry updates (Sect. 3) were documented with simulations
covering summer when secondary HCHO is highest.


Simulated HCHO is highest in the southeastern US (Fig. 2) in the summer (Fig. S9) due to secondary HCHO from biogenic
emissions and photochemical activity. High levels of HCHO are also simulated in California in forested areas surrounding the
Central Valley. HCHO in CRACMM2 is higher compared to CRACMM1 in most areas, with the largest increases in summer,
though there are some places with seasonal reductions in HCHO of up to -0.1 ppb. Besides the southeastern US and in parts
of California where biogenic emissions of isoprene are highest, summer HCHO is also increased across the eastern US broadly.
Changes in HCHO in the western US (outside of California) are small (<0.2 ppb). CRACMM2 simulates increased HCHO in
the summer across the boreal forests of Canada and forested areas of Mexico within the modeling domain. Predicted spring
and fall HCHO also increases in CRACMM2 for the eastern US, California, and Mexico, but to a lesser degree than in the
summer (Fig. S9). Overall, 2019 June–August surface HCHO during peak photochemical production (11am–3pm) is increased
by 0.6 ppb (32%) over the southeastern US and by 0.2 ppb (13%) over the entire CONUS.



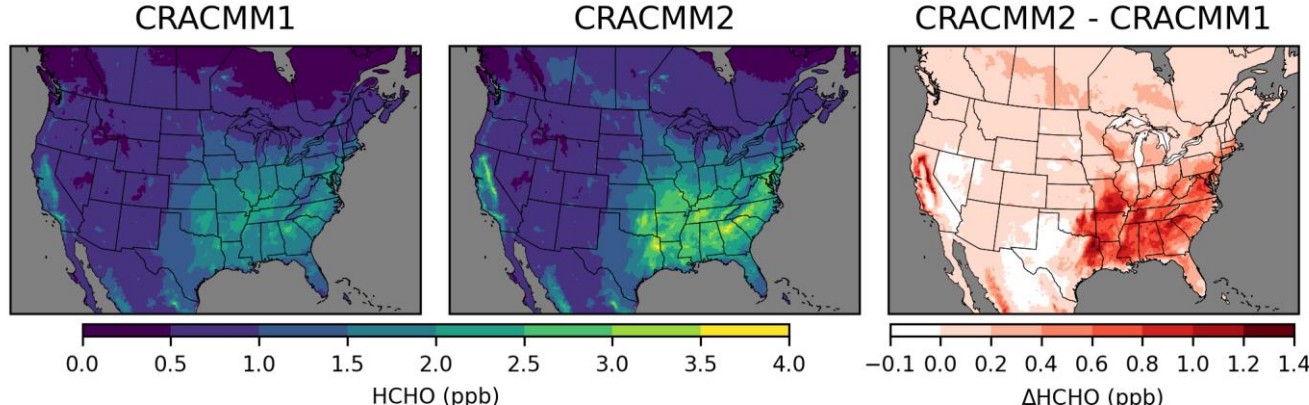

**Figure 2. Surface layer 11 am–3 pm local time June–August 2019 average HCHO concentrations simulated with CRACMM1 (left) and CRACMM2 (middle) and the change in CRACMM2 compared to CRACMM1 (right). Analogous results for other seasons are provided in Fig. S9.**

Chemistry updates were implemented in stages to track the incremental effects of updates to different chemical systems (Fig. 3; Fig. S10). The update of isoprene chemistry to the AMOREv1.2 isoprene condensation from the RACM2-based isoprene chemistry of CRACMM1 had by far the largest impact on HCHO, and the impacts of the isoprene updates dominate the difference in HCHO between CRACMM1 and CRACMM2. HCHO concentrations most dramatically increase in the southeastern U.S. where biogenic emissions, dominated by isoprene, are highest. Widespread increases in HCHO of ~0.5 ppb occur throughout much of the rest of the eastern US and the boreal forests of Canada as a result of the increased isoprene HCHO yields. After isoprene, the monoterpene chemistry updates had the largest impact on HCHO, accounting for ~10% of the total increase in HCHO in CRACMM2 compared to CRACMM1. The impacts on HCHO are spatially representative of biogenic monoterpene emissions with the largest increases in the southeastern US and smaller increases extending to much of the rest of the eastern US. On the west coast, monoterpene impacts have a different spatial pattern than was seen for the isoprene updates as the forests in the Pacific Northwest have larger fraction of total biogenic emissions from monoterpenes compared to the southeastern US.



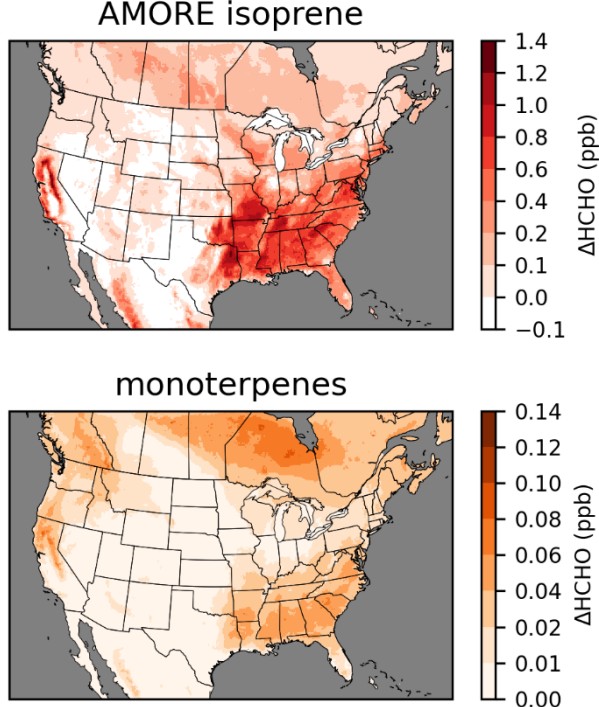

**Figure 3. Incremental impacts on surface layer 11 am–3 pm local time 2019 June–August average HCHO resulting from AMORE isoprene chemistry updates and monoterpene chemistry updates. Incremental impacts of other chemistry updates are provided in Fig. S10.**

Effects on HCHO from other CRACMM2 chemistry updates are small in comparison to the isoprene and monoterpene updates (Fig. S10). The inclusion of ECH4 results in some localized increases in HCHO near extremely large methane sources. ECH4 emissions included in CMAQ here do not include all methane emissions that are available from the gridded EPA U.S. methane greenhouse gas inventory (Maasakkers et al., 2023) but only include emissions for traditional NEI sources. The updated heterogeneous chemistry results in small (<40 ppt) increases in HCHO in the southeastern US due to decreased $HO_X$ from uptake of $HO_2$ marginally increasing the lifetime of HCHO. The aromatic chemistry updates result in small (<10 ppt) increases in HCHO which are localized to areas with high styrene emissions. More detail on the effects on HCHO from these updates is given in the SI and Fig. S10.

Many of the updates in CRACMM2 have been targeted at secondary HCHO, but the updates also affect $PM_{2.5}$ and $O_3$. Since $PM_{2.5}$ and $O_3$ are not the focus of this work, we provide only a brief overview here. Many more details on $PM_{2.5}$ and $O_3$ impacts are documented in the SI for interested readers. $PM_{2.5}$ decreased across the CONUS in CRACMM2 compared to CRACMM1. For $PM_{2.5}$, the annual mean bias across sites in the Air Quality System (AQS) database went from -0.5 µg m$^{-3}$ in CRACMM1 to -0.8 µg m$^{-3}$ in CRACMM2 driven by reductions in organic aerosol in CRACMM2 from reduced HOM formation from




monoterpene nitrates in CRACMM2 (Sect. 3.4). These decreases are partially offset by new SOA pathways through
heterogeneous uptake of isoprene-derived compounds (Sect. 3.1 and 3.3). The changes improve the performance of organic
carbon which is biased high for the annual average (both in CRACMM1 and in CRACMM2). Low biases in $PM_{2.5}$ mass come
from low biases in other $PM_{2.5}$ species including sulfate (Vannucci et al., 2024), nitrate, ammonium, and elemental carbon.
Annual average max daily 8-h average (MDA8) $O_3$ increased in CRACMM2 in the eastern US (particularly in the southeastern
US) and in California. MDA8 $O_3$ decreased slightly (<0.5 ppb) in western Texas and throughout the central US. The changes
in $O_3$ come primarily from changes in $HO_X$ resulting from the implementation of the AMORE isoprene chemistry condensation
and from increased $NO_X$ recycling from monoterpene nitrates. Annual mean bias in MDA8 $O_3$ across AQS sites improved
from -1.1 ppb in CRACMM1 to -0.7 ppb in CRACMM2, though there are spatial and seasonal differences in biases that offset
each other. On average across all sites, underestimates in MDA8 $O_3$ in the spring improve in CRACMM2 and a high bias in
summer to early fall MDA8 $O_3$ becomes slightly worse in CRACMM2.

## 5 Comparisons to observations

CMAQ HCHO results are compared against several different sources of observations to evaluate the impacts of the
CRACMM2 updates. Observational data includes satellite based-observations from TROPOMI, aircraft-based observations
from the Fire Influence on Regional to Global Environments and Air Quality (FIREX-AQ) campaign, and surface-level hourly
observations.

### 5.1 TROPOMI

TROPOMI onboard the Sentinel-5 Precursor satellite provides once daily coverage at around 13:30 local solar time. We use
the TROPOMI HCHO tropospheric vertical column density (VCD) and compare with the HCHO VCD simulated by CMAQ.
TROPOMI and CMAQ data are processed for comparison using the cmaqsatproc python tool
(https://github.com/barronh/cmaqsatproc). We use a reprocessed TROPOMI HCHO dataset with a resolution of 5.5 km × 3.5
km which uses version 2 of the level 2 processor for all of 2019. TROPOMI data are filtered to include only data with a quality
assurance (QA) value > 0.75 (stricter than the QA value > 0.5 recommended minimum). A QA value > 0.5 indicates no error
flag, cloud radiance fraction at 340 nm < 0.5, solar zenith angle <= 70°, surface albedo <= 0.2, no snow/ice warning, and air
mass factor (AMF) > 0.1 (KNMI, 2023). TROPOMI data are gridded onto the 12 km × 12 km CMAQ model grid and are
updated with an AMF based on the CMAQ HCHO vertical profile. For each comparison of a CMAQ simulation to TROPOMI,
the AMF derived from that specific CMAQ simulation is used. The daily TROPOMI HCHO VCDs are scaled up by 25%
when the HCHO VCD exceeds $8×10^{15}$ molecules cm$^{-2}$ to account for a low bias in TROPOMI HCHO at high HCHO VCD
levels (De Smedt et al., 2021) and then averaged seasonally. The CMAQ data are sampled so that CMAQ VCDs are only
retained for model grid cells and days when there is valid TROPOMI data.



The summer average HCHO VCD from CMAQ (with CRACMM1 and CRACMM2) broadly reproduces the spatial distribution of TROPOMI, with the highest HCHO occuring in the southeastern US along with another area of high HCHO surrounding the Central Valley of California (Fig. 4, Fig. S11). The updates introduced in CRACMM2 increase column HCHO in the eastern US, particularly in the southeastern US, and in California. These increases are mostly from increased HCHO from isoprene from biogenic emissions with some additional increases from monoterpene HCHO yields which are also mostly

from biogenic sources. CMAQ becomes closer to TROPOMI with these increases, though HCHO is still consistently lower than TROPOMI throughout the CONUS. In most areas, however, the HCHO VCD simulated by CMAQ is within the range of TROPOMI uncertainty (Fig. S12). The largest underestimates in HCHO occur in the western US. HCHO is significantly underestimated in the Permian Basin, a major oil and gas producing area in western Texas and New Mexico. HCHO is also underestimated over other oil producing areas in Texas and Oklahoma, specifically over the Ft. Worth and Anadarko Basins

which could be due to underestimates in primary HCHO, other ROC precursor emissions, and/or secondary production. CTM simulations with WRF-Chem using the fuel-based inventory of oil and gas (FOG) (Gorchov Negron et al., 2018; Francoeur et al., 2021) showed higher HCHO VCDs over the Permian Basin than our simulations here (Dix et al., 2023). Comparisons of FOG to the 2014 NEI have shown that FOG had lower $NO_X$ emissions and higher non-methane VOC emissions (Francoeur et al., 2021). The emissions inventory used in our simulations is based on the 2017 NEI with some updates (see EQUATES,

Foley et al. (2023)), and more recent versions of the NEI may show different results. Emissions of both $NO_X$ and ROC precursors will both affect HCHO production in this area (Dix et al., 2023). A sensitivity simulation in which $NO_X$ and ROC emissions from oil and gas sources were doubled resulted in increases in summer average HCHO VCD at the TROPOMI overpass time of up to $1.4\times10^{15}$ molecules cm$^{-2}$ and increases of surface level 11am–3pm summer average HCHO of up to 0.5 ppb (Fig. S13).


Comparison to the TROPOMI column HCHO indicates some regional biases in CMAQ. TROPOMI column HCHO is consistently higher than CMAQ values in the Mountain West and the southwestern US. A large underestimate is seen in Arizona over the Tonto National Forest to the northeast of Phoenix. Large underestimates in California occur over the Los Angeles metropolitan area and over national forest land east of the Central Valley. Underestimates over the national forest

land in Arizona and California could result from underestimated biogenic emissions. The underestimated HCHO in Los Angeles is more likely related to anthropogenic precursors and could result from either underestimated precursor emissions or secondary production. More detailed data and analysis of these individual areas, such as might be possible with a field campaign, are likely needed to explore the specific reasons for the underestimates of HCHO. In the part of the modeling domain covering Canada, CMAQ HCHO is consistently higher than TROPOMI. HCHO is extremely overestimated (by >$10^{16}$

molecules cm$^{-2}$) by CMAQ in parts of Manitoba and Ontario due to excessive primary HCHO from wildfires which likely resulted from inaccurate representation of the emissions and/or plume trajectories from these fires in the model. Updates to HCHO production in CRACMM2 increase the HCHO VCD in the eastern US by ~$1\times10^{15}$ molecules cm$^{-2}$ on average with increases of up to ~$4\times10^{15}$ molecules cm$^{-2}$ in the southeastern US, leading to a better comparison with TROPOMI HCHO.



However, several additional areas with underestimated HCHO (e.g., the Permian Basin and parts of Arizona and California)
still need more exploration in future work. More detailed analysis is needed to understand the roles of precursor emissions,
secondary HCHO production, and the diurnal variability of HCHO as compared to observations.

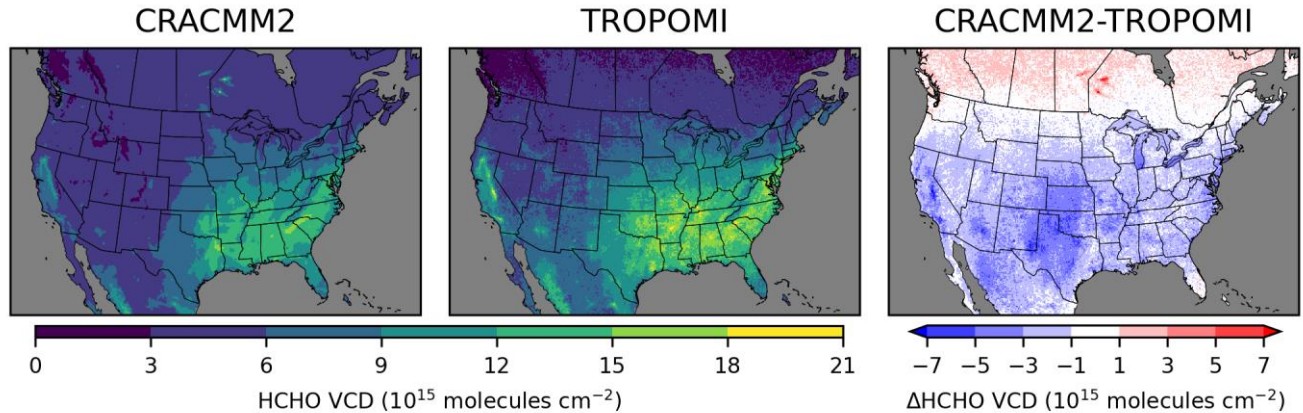

**Figure 4. June–August 2019 average tropospheric vertical column densities from CMAQ with CRACMM2 (left) and from**
**TROPOMI (middle) and the difference between CRACMM2 and TROPOMI (right). Similar comparisons for other seasons are**
**provided in Fig. S11.**

## 5.2 FIREX

As part of the FIREX-AQ experiment, in-situ measurements of HCHO (among many other trace gas and aerosol
measurements) were taken to assess the chemical evolution of fire plumes by sampling from the NASA DC-8 aircraft during
the summer of 2019 (Liao et al., 2021; Warneke et al., 2023). While FIREX-AQ was targeted towards fires, measurements
also include conditions outside of wildfire plumes. A significant amount of data was collected outside of fire plumes and is
more representative of background conditions than fire conditions. We use HCHO data from two instruments onboard the DC-
8 during FIREX-AQ. One is the In Situ Airborne Formaldehyde (ISAF) instrument (Cazorla et al., 2015) which uses laser-
495  induced fluorescence to measure HCHO. The second is the Compact Atmospheric Multispecies Spectrometer (CAMS)
(Richter et al., 2015) which is a mid-IR laser-based spectrometer. During FIREX-AQ, HCHO measured by the ISAF and
CAMS instruments were highly correlated with an $r^2$ of 0.99 and an intercept near zero but with a slope of 1.27 based on an
orthogonal regression between the two. Follow up studies indicated that this discrepancy was due to differences in the
calibration standards employed (Liao et al., 2021). We include both the ISAF and CAMS observations in our analysis and
500  interpret their difference as an indicator of measurement uncertainty. FIREX-AQ observations at 1 Hz frequency were averaged
up to the minute and were paired with the CMAQ model outputs coincident in space and time with the flight track by matching
the observation time to the nearest hourly model output time step, the radar altitude to the model vertical layer height, and the




aircraft coordinates to the corresponding model horizontal grid cell. Paired observation-model data are then separated into "smoke" or "background" categories based on a smoke indicator flag which is based on CO and black carbon enhancements above background concentrations. Starting from a total of 9084 paired data points available, 7568 (83%) had measurements available for both ISAF and CAMS HCHO. Of these, 1932 (26%) were flagged as smoke with the remaining 5636 (74%) taken as background.

We focus on the background (i.e., not in fire plumes) data since HCHO from fires and within fire plumes was not a focus of the CRACMM2 updates. (See Pye et al. in prep. for an evaluation of CMAQ-CRACMM1AMORE predictions of HCHO during FIREX-AQ.) Although these data are sampled outside of fire plumes, there still may be some influences from fire emissions even in the background observations since the data are collected in fire-affected regions during periods with active fires. Data are further separated geographically to highlight differences in CMAQ performance in California, the rest of the western US, and the eastern US with a longitude of -97 °W defining the east-west boundary. The data in California primarily sample the Central Valley and the Los Angeles area. The data in the rest of the western US sample within the states of Arizona, Idaho, Utah, Washington, and Montana. The data in the eastern US are exclusively in the southeastern US. The vertical profile of HCHO in CMAQ is evaluated with the FIREX-AQ HCHO measurements (Fig. 5). Data are aggregated by altitude in bins of 200 m below 3000 m and 500 m above 3000 m to generate a campaign average HCHO vertical profile in each geographic region. Across all regions that were sampled during FIREX-AQ, the simulated vertical profile of HCHO follows the basic shape of the observations with the highest values in the boundary layer and decreases with altitude. Above ~2 km, the CMAQ vertical profile is biased low across all regions, and the CRACMM2 updates have negligible effects. The modeled near-surface concentrations are very low in California (1-2 ppb below observations depending on the instrument). The low bias in HCHO aloft may be from underestimated precursor abundance aloft and/or from underestimated secondary production from the dominant aloft precursors. The low bias in HCHO aloft may also explain some of the low biases in HCHO VCDs from CMAQ compared to TROPOMI (Sect. 5.1) since TROPOMI has a greater sensitivity at higher altitudes. Near-surface HCHO is also biased low in the rest of the western US, though with a smaller magnitude. The updates in CRACMM2 have only small effects even near the surface in the western US. In the southeastern US, however, CRACMM2 updates lead to an increase in HCHO below 2 km which improves the low bias in CRACMM1. The CRACMM2 southeastern US predictions at lower altitudes are consistent with measurements as they fall between the ISAF and CAMS measurements. The CRACMM2 updates primarily affect secondary HCHO from biogenic emissions, so increases in HCHO in the southeastern US are expected and are consistent with the impacts shown in previous sections.





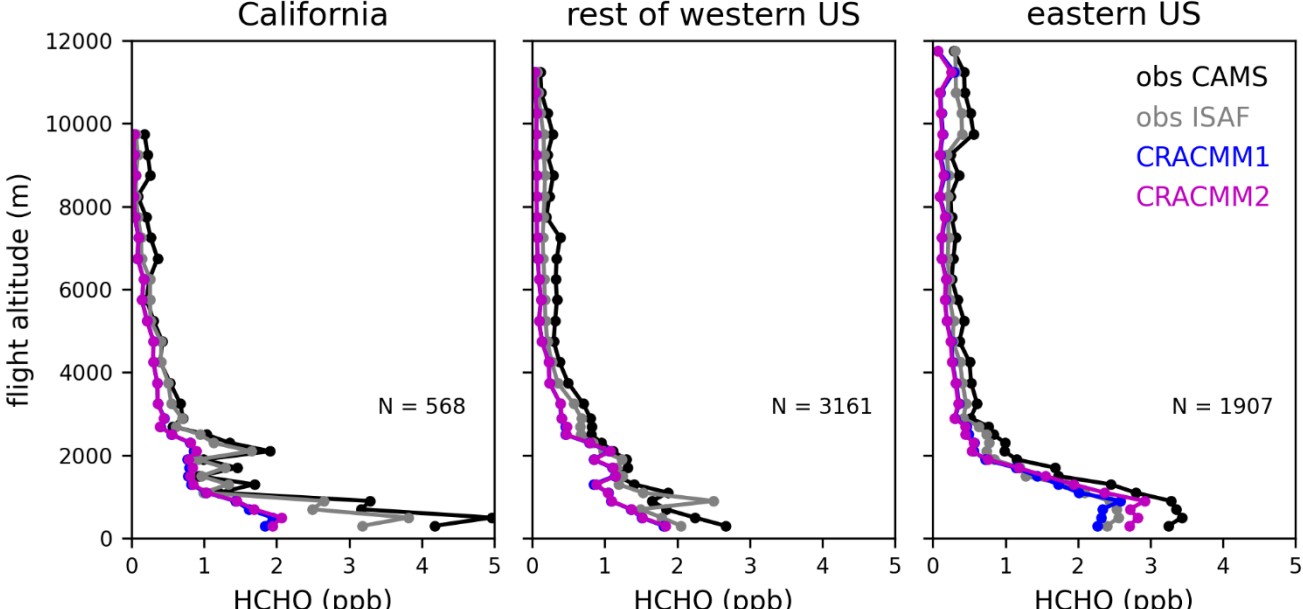

**Figure 5. FIREX-AQ campaign average vertical profiles of observed (CAMS and ISAF) and simulated (CRACMM1 and CRACMM2) HCHO. Data flagged as within smoke plumes is excluded here. Profiles are separated into western and eastern US using a longitude of -97 °W. Data over California is further separated from the rest of the western US. The vertical profiles show the average HCHO over altitude bins of 200 m below 3000 m and 500 m above 3000 m. The number of observations (N) in each geographical area is also provided.**

### 5.3 Hourly surface observations

HCHO observations from federal, state, local, and tribal air quality monitoring networks are available from the AQS database. Many HCHO observations from AQS are based on a 24-h sample collection (i.e., daily average) with offline characterization (method TO-11A), though some sites collect 8-h samples over the course of a day on a once per three days schedule during the summer. The lack of hourly data for evaluation of the HCHO diurnal variability in CMAQ is a limitation of the AQS HCHO observations. In addition, previous work indicates offline network measurements of HCHO can be biased high or low (Zhu et al., 2017; Mouat et al., 2024), and we find AQS measurements show a summer, regional maximum in HCHO in the Carolinas (Fig. S17-S18) rather than in the northern Georgia region, in contrast to CMAQ and TROPOMI (Fig. 2 and 4). Here, we focus on surface HCHO observations with hourly resolution from episodic field intensives to better understand drivers of concentrations. In several cases, due to data limitations, we leverage observations from a year other than our 2019 modeling year. As temperature is a strong driver of isoprene emissions and can modulate chemistry, some deviation between the model predictions and observational data is expected (more analysis of HCHO variation with temperature is provided in Fig. S20-S21). Rather than evaluating the performance of the hourly HCHO in CMAQ quantitatively, we use the hourly measurements available in other years as a qualitative indication of how well CRACMM2 in CMAQ represents the typical diurnal variability



of HCHO. Data are paired by hour and date across observed and modeled years, and hourly data points with missing observations are dropped before averaging to the diurnal cycle. Comparisons with routine AQS data are available in the supplement, and details on sampling locations, dates, and instrumentation used here are provided in Table S4.

The Salt Lake City, UT, data from winter 2017 covers periods with persistent cold-air pool (PCAP) events which are

characterized by extremely shallow mixed layers that prevent vertical mixing. These events are often not well-captured by meteorological models that drive CTMs, so we exclude data collected during three PCAP events (13-20 January, 27 January - 4 February, and 13-18 February). The Salt Lake City observations show a relatively flat diurnal profile with slight peaks in the late morning and in the evening (Fig. 6). The CRACMM diurnal profile is also flat with small peaks in the morning and in the early afternoon. The magnitude of the simulated HCHO diurnal profile is lower by about a factor of 2.5 on average compared

to the observations. Although the comparison uses different observation and simulation years, it suggests a missing anthropogenic source in the model emission inventory since biogenic emissions would not be a major factor during the winter sampling period. Previous work suggested primary HCHO emissions are underestimated in the Salt Lake City area based on data collected during the Salt Lake Regional Smoke, Ozone and Aerosol Study (SAMOZA) campaign in summer 2022 (Ninneman et al., 2023; Jaffe et al., 2024). Primary HCHO is expected to contribute relatively more to overall HCHO in the

winter as compared to warmer seasons due to the longer lifetime of HCHO in the winter and reduced biogenic precursor emissions. Model simulations have estimated primary HCHO fractions in the winter of 25-50% (Luecken et al., 2012). Secondary production is still important in winter, and photochemistry can be enhanced through increased albedo in snow-covered areas (Edwards et al., 2014). While this data is suggestive of underestimated anthropogenic emissions in the area, the missing driver cannot be identified beyond a combination of primary HCHO and/or ROC precursors.


For the several locations in the northeastern US (Westport, New Brunswick, and Flax Pond) in summer (see Fig. S22-S23 for other seasons), the comparisons of continuously sampling online techniques (in 2023) to simulation predictions (in 2019) are generally consistent and indicate the model captures the correct broad features of HCHO. The simulated HCHO reaches about the same midday peak level as the observations when the CRACMM2 updates are added. While the model does reflect a

daytime increase in HCHO at these sites, the simulated diurnal profile shows less diurnal variation than the observations. The observations show a sharp rise from the early morning to a midday peak, followed by a sharp decline over the late afternoon and into the night. The comparisons of observed diurnal variability of HCHO with CRACMM1 and CRACMM2 in CMAQ indicate HCHO in CMAQ tends to be too high at night.

The error in the HCHO diurnal profile during summer in CMAQ is pronounced for two sites in Atlanta, GA, where data has been collected as part of a longer-term HCHO sampling effort (Mouat et al., 2024). One site is co-located with a Photochemical Assessment Monitoring Stations (PAMS) network site, known as the South DeKalb (SDK) monitoring site, located in a suburban part of the Atlanta metro area. The other site is located on the campus of Georgia Tech (GT) which is within the





urban core of the city of Atlanta. The two Atlanta sites are located ~15 km away from one another and are in adjacent grid

cells of the 12 km CMAQ modeling domain. At both Atlanta sites, the observed diurnal profile begins increasing at 6 am until

it reaches peak levels around 11 am to 3 pm before dropping again into the late afternoon and overnight (Fig. 6). The overnight

lows at the SDK site are lower than at the GT site, though the diurnal variation (i.e., the difference between the high and low

values) at each site is similar. The modeled diurnal profile does not reproduce the observed shape at either site. The model

correctly reflects the start of the rise in HCHO at 6 am; however, predicted HCHO in CRACMM1 and CRACMM2 declines

in the late morning, remains flat as the afternoon progresses, then has a slight rise at night. Similar discrepancies occur for

other seasons (Fig. S23). Across seasons, CMAQ does not capture the peak HCHO during midday for several possible reasons.

Biogenic isoprene emissions could be low in CMAQ. The observed HCHO diurnal profile largely follows the typical daily

cycle of isoprene emissions, and secondary HCHO from isoprene is expected to be the dominant contributor to HCHO in the

southeastern US. A comparison of the modeled diurnal profile of isoprene in 2019 to observations in 2022 and 2023 (hourly

isoprene measurements are not available at the SDK site for 2019) shows that simulated isoprene is within the range of

interannual variability (Fig. S24). However, the 2019 simulated isoprene diurnal profile decreases between noon and 5 pm

whereas the observed isoprene in 2022 and 2023 continues to increase or remains near its peak during this period. Besides

isoprene, another potential contributing factor to the low midday HCHO could be that the loss rate of HCHO is too high so

that HCHO is lost faster that it can be produced, contributing to the lack of peak during midday. For instance, if cloud coverage

is underestimated in the model, the photolysis losses could be too high. In all seasons except winter (when HCHO is very low

at all times), the modeled Atlanta nighttime values are typically higher than the observations, especially after the CRACMM2

updates. The high nocturnal HCHO does not seem to result from a shallow modeled boundary layer. Modeled CO (used here

as an indicator for boundary layer depth) decreases at night while observed CO increases, indicating that the modeled boundary

layer is too deep rather than too shallow (Fig. S24).




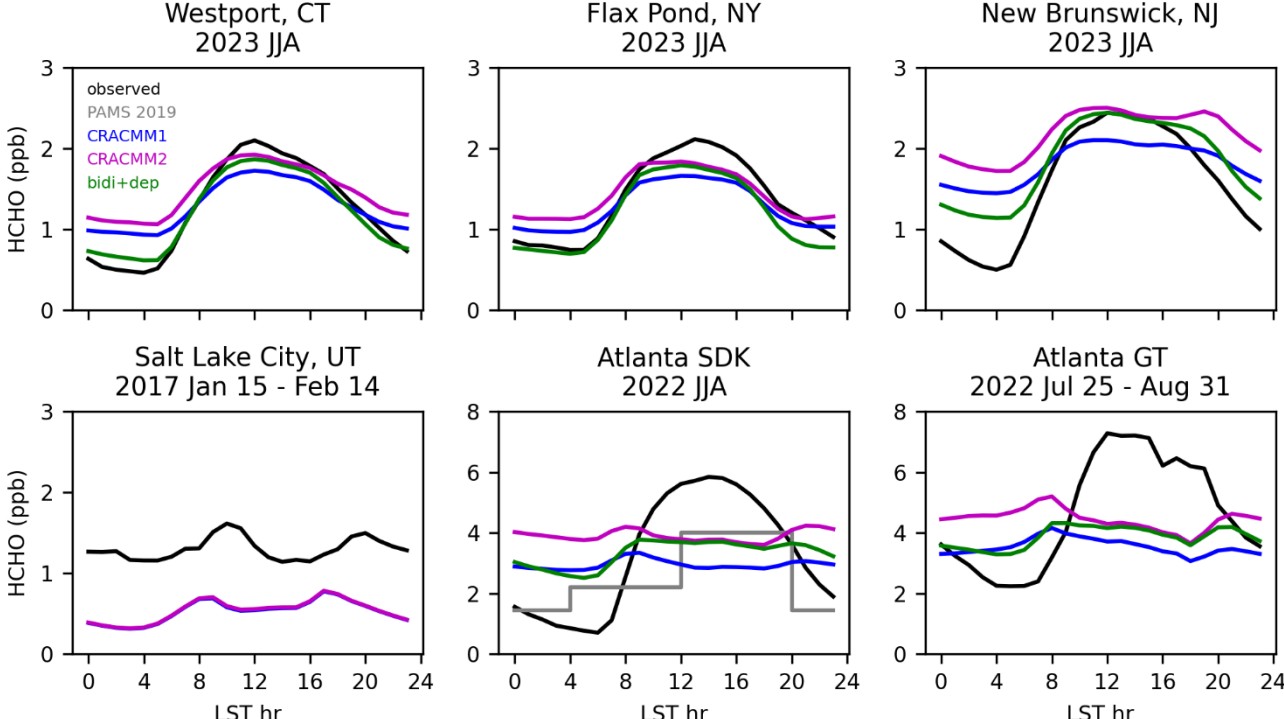

**Figure 6. Diurnal profiles of observations in several years at several sites compared to CMAQ simulations in 2019 using CRACMM1, CRACMM2, and CRACMM2 with updated HCHO bidirectional flux and deposition (bidi+dep). Sampling locations and dates are provided above each panel. PAMS 2019 (grey line) shows the average of 8-h HCHO samples collected using method TO-11A during summer 2019 at the SDK monitoring site.**

## 5.4 Deposition updates

HCHO is expected to decline at night, as is seen in the hourly observations, since HCHO production is primarily driven by photochemistry. The consistently high predicted nighttime HCHO levels compared with observations from multiple locations suggest a missing nighttime loss process for HCHO in the model. Bidirectional exchange of HCHO on plant surfaces has been proposed and measured in a laboratory setting (Shutter et al., 2024). Bidirectional exchange of formic acid has also been previously implemented in CMAQ, resulting in improvement of the diurnal variability from a previously flat modeled diurnal profile to one more consistent with surface observations (Gao et al., 2022). We performed a sensitivity simulation for summer 2019 where the STAGE dry deposition model in CMAQ was updated to add a bidirectional flux for HCHO based on the HCHO compensation point parameterization of Shutter et al. (2024) and a relative humidity (RH) dependence to leaf wetness for dry deposition (Altimir et al., 2006; Burkhardt et al., 2009).





These updates to deposition lead to better agreement of the modeled and observed diurnal profiles (Fig. 6). The addition of the bidirectional flux of HCHO tends to slightly increase HCHO throughout all hours of the day. The leaf wetness deposition tends to reduce HCHO throughout all hours of the day with smaller decreases during the day and larger decreases at night, consistent with the typical diurnal variability of RH which is higher at night. The increased HCHO from the bidirectional flux mostly offsets the increased deposition losses during the day. At night, the increase in deposition reduces HCHO, leading to better agreement with nighttime observations. For two of the northeastern US sites (Westport and Flax Pond), the HCHO at night becomes very close to the observations after the deposition updates are added (Fig. 6). At the New Brunswick site, HCHO is reduced at night which better matches observations but is still higher than observed. For the two Atlanta sites, the addition of the bidirectional flux of HCHO and the increased deposition leads to better agreement with the observed diurnal profile. The shape of the diurnal profile becomes more like the observations, falling at night and peaking during the day. However, the model still does not quite capture the lows at night, particularly at the SDK site, or the height of the peak during midday. The bidirectional flux and deposition updates slightly reduce surface and column HCHO by up to 0.15 ppb (June–August 11 am–3 pm average) and $0.3 \times 10^{15}$ molecules cm$^{-2}$ (June–August average at TROPOMI overpass) (Fig. S25). The June–August nocturnal (8pm–4am) surface HCHO is reduced on average by 1.1 ppb (36%) over the southeastern US and 0.5 ppb (29%) over the entire contiguous US.

## 6 Implications

The increased HCHO in CRACMM2 has implications for the estimation of cancer risk as HCHO is a leading driver of cancer risk from ambient exposure to HAPs (Strum and Scheffe, 2016). A significant amount of HCHO originates from oxidation of biogenic ROC, primarily isoprene. However, anthropogenic emissions of ROC precursors also contribute to HCHO, and anthropogenic NO$_X$ affects the secondary production of HCHO (Valin et al., 2016; Wolfe et al., 2016b). Here, we estimate a controllable fraction of HCHO and its resulting cancer risk by performing an annual simulation where anthropogenic emissions of NO$_X$ and ROC (excluding anthropogenic fire emissions) within our 12 km CMAQ modeling domain are set to zero. Using the anthropogenic zero out simulation, we estimate the controllable fraction of HCHO simulated in CMAQ with CRACMM2 as the HCHO concentration in the CRACMM2 base simulation minus the HCHO concentration in the zero anthropogenic simulation divided by the HCHO concentration in the base simulation (Fig. 7). The controllable portion of the annual average over land is 24% on average and ranges from 2% to 97% over the CONUS. Seasonally, the highest controllable fraction occurs in winter (average over land of 46%), particularly in the northern portions of the domain (Fig. S26) consistent with increased primary HCHO from residential heating along with longer HCHO lifetimes and reduction in biogenic emissions in winter. The controllable fraction is lowest in the summer (average over land of 17%) when photochemistry is most active, biogenic precursors are highest, and HCHO concentrations are at their highest. The lifetime of HCHO against photolysis is also shortest during this time which limits the impact of primary HCHO. Here we define controllable to include anthropogenic emissions of short-lived precursors NO$_x$ and ROC, but this definition neglects the effects of global background methane oxidation on



HCHO. Methane has more than doubled in concentration since the preindustrial era and has a lifetime of ~12 years (Prather et al., 2012), such that reductions in methane could impact HCHO concentrations over large spatial scales in the near term. Future work may consider the role of methane in the fraction of controllable HCHO.

The increased cancer risk from a lifetime of exposure to ambient HCHO is estimated as the annual average concentration times

the unit risk estimate (URE). The URE of HCHO of $1.3\times10^{-5}$ (µg m$^{-3}$)$^{-1}$ indicates 13 more people might be expected to develop cancer per one million people exposed daily for a lifetime to 1 µg m$^{-3}$ of HCHO. For purposes of estimating risk, we apply an assumed lifetime of exposure of 70 years to our predicted annual average concentrations. The gridded cancer risk estimate is used along with 2019 American Community Survey (ACS) block group level population estimates which are gridded onto the 12 km model domain to calculate the CONUS population-weighted cancer risk and to make an estimate of the total number of

CONUS cancer cases estimated from HCHO. Cancer cases are calculated as the sum over CONUS grid cells of the gridded cancer risk times the gridded population (equivalently: the population-weighted cancer risk over CONUS grid cells times the CONUS population). The population-weighted cancer risk (not mortality) for exposure to HCHO in ambient air predicted by CMAQ increases from 17 in a million with CRACMM1 to 19 in a million with CRACMM2, of which 8 in a million (~40%) is estimated to be controllable. The estimate of CONUS cancer cases increases from 5400 with CRACMM1 to 6200 in

CRACMM2, of which 2500 are estimated to be controllable (Fig. 7). For reference, the national average risk from exposure to ambient HCHO from the 2019 AirToxScreen assessment implies a lifetime risk of ~4800 cancer cases (using the ACS 2019 population estimate). While the results from AirToxScreen are typically rounded to one significant digit, we retain two significant digits here to better compare results from different simulations. Some differences between AirToxScreen and this work are expected given differences in the CMAQ model version (5.3.2 in AirToxScreen vs. 5.4 here), the chemical mechanism

(cb6r3 in AirToxScreen vs. CRACMM1 and CRACMM2 here), the WRF version (3.8 in AirToxScreen vs. 4.1.1 here), the baseline anthropogenic emissions inventory (2017 NEI in AirToxScreen vs. EQUATES emissions here), and the use of a hybrid CTM and dispersion model approach in AirToxScreen vs. CTM results alone here.





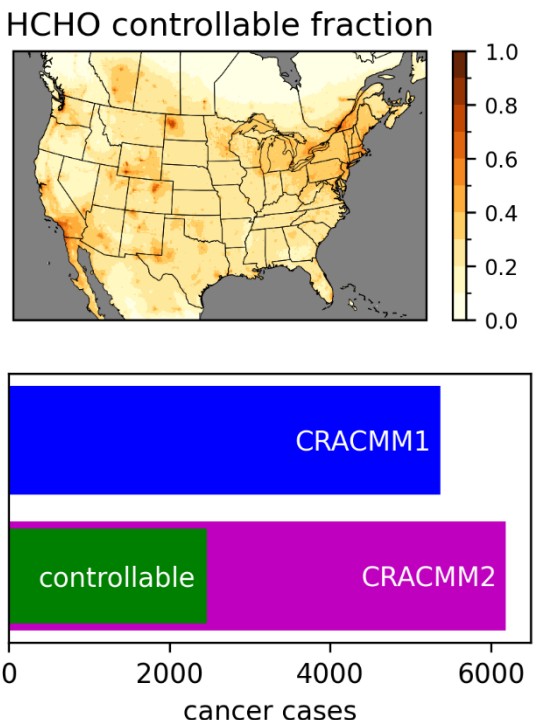



In this work, an updated representation of deposition and secondary production of HCHO improves our ability to simulate ambient HCHO and its consistency with observations from satellite remote sensing, FIREX-AQ field data, and hourly surface measurements. The investigation here and upcoming data indicate avenues for future work to further improve our understanding of drivers of ambient concentrations. For example, comparison of the diurnal variability of HCHO against hourly surface observations showed that CRACMM2 was typically too high at night, pointing to the potential for a missing nighttime loss pathway for HCHO in CMAQ. The ability of nocturnal leaf wetness to modulate dry deposition and therefore abundance of HCHO suggests concentrations of other soluble species could also be improved by updates to dry deposition or bidirectional exchange. In CRACMM2, peak HCHO levels were near observed levels for surface sites in the northeastern US; however, for daytime in the southeastern US and across the free troposphere, values in CRACMM2 were lower than observed. This suggests improvements to precursor abundance and/or secondary production is still needed. More in-depth explorations of HCHO and its precursors may be possible with data from the 2023 AGES+ field campaigns (https://csl.noaa.gov/projects/ages) and with the new geostationary satellite-based HCHO data from Tropospheric Emissions:





Monitoring of Pollution (TEMPO) mission (https://tempo.si.edu) which will provide daytime variation in HCHO and could enable further improvements in HCHO and its precursors in CRACMM. In addition, we focused exclusively on ambient air in this work, but indoor air concentrations of HCHO can be substantial (Salthammer et al., 2010). A more complete representation of inhaled HCHO health risk will require further improvements to predictions for ambient air as well as characterizing exposure for the indoor environment and extending this analysis to health endpoints beyond cancer.

**Code and data availability**

- The CMAQ source code is available from GitHub (github.com/USEPA/CMAQ) and Zenodo (https://doi.org/10.5281/zenodo.7218076).
- The CRACMM GitHub site (github.com/USEPA/CRACMM) provides files needed to run CRACMM2 in F0AM, the complete CRACMM2 mechanism, and CRACMM2 species descriptions and properties.
- The F0AM code is available from GitHub (github.com/AirChem/F0AM).
- FIREX-AQ observational data are available from the FIREX-AQ data archive (https://www-air.larc.nasa.gov/cgi-bin/ArcView/firexaq). CAMS HCHO data is revision R3. ISAF HCHO data is revision R0. Navigational data is revision R1.
- HCHO observational data for Atlanta are available from GitHub (github.com/KaiserLab-GeorgiaTech/long-term-HCHO-monitoring_efforts_datasets) and Zenodo (https://doi.org/10.5281/zenodo.10855090).
- HCHO observational data for summer 2023 at Westport, Flax Pond, and New Brunswick sites are available from the following data archive: https://www-air.larc.nasa.gov/cgi-bin/ArcView/listos.2023. Data from all three sites are revision R0.
- HCHO observational data for winter 2017 in Salt Lake City are available from the following data archive: https://csl.noaa.gov/groups/csl7/measurements/2017uwfps/Ground/DataDownload/. Data is revision R0.
- Additional supporting data will be available at data.gov upon publication of the final manuscript.

**Author contributions**

TNS performed all simulations and analyses and wrote the initial draft. TNS and HOTP designed the research. TNS, HOTP, ELD, RHS, and IRP developed monoterpene chemistry. FCW and VFM developed the AMORE isoprene chemistry in consultation with HOTP and TNS. HOTP, BHH, JOB, and BKP provided additional code and analysis. ARW, APM, JK, GMW, JMS, TFH, and AF provided ambient data. All coauthors contributed to reviewing and editing the manuscript.



**Acknowledgements**

This work was supported by the U.S. Environmental Protection Agency Office of Research and Development. This research was supported in part by an appointment to the U.S. Environmental Protection Agency (EPA) Research Participation Program administered by the ORISE through an interagency agreement between the U.S. DOE and the U.S. Environmental Protection Agency. ORISE is managed by ORAU under DOE contract number DE-SC0014664. The views expressed in this paper are

those of the authors and do not necessarily represent the views or policies of the U.S. Environmental Protection Agency, the U.S. DOE, or ORISE. We thank TROPOMI and AQS teams for providing data and the CMAQ team for additional discussion. We thank Golam Sarwar and Doris Chen for comments on a draft version of the paper. GMW, JMS, and TFH acknowledge support from the NASA Tropospheric Composition Program and NOAA Climate Program Office's Atmospheric Chemistry, Carbon Cycle and Climate (AC4) program (NA17OAR4310004).

**Disclaimer**

The authors have no competing interests to declare.

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
