# Peer review of "Role of chemical production and depositional losses on formaldehyde in the Community Regional Atmospheric Chemistry Multiphase Mechanism (CRACMM)"

_EGUsphere, 2024_

## Author Comment (AC1)

We would like to thank the reviewers for their comments. Responses to reviewer comments are provided below in blue, and new text added to the manuscript is provided in red.

**Reviewer 1 (J.-F. Müller):**

This manuscript presents a comprehensive revision of the chemical mechanism used in the CMAQ model, the CRACMM mechanism. The goal is to achieve a better representation of the chemical production of formaldehyde from its various VOC precursors in CMAQ, mainly motivated by the significance of atmospheric formaldehyde for human health. The paper also presents a detailed evaluation of CMAQ model predictions against in situ, airborne (FIREX) and satellite (TROPOMI) formaldehyde data, and shows that the mechanistic updates generally improve the model performance. A consequence of the updates is an enhancement of formaldehyde abundances, especially over the regions with high biogenic emissions like the southeast US. Next, the model is used to estimate the number of cancer cases due to airborne HCHO in the United States, as well as the controllable fraction of this number.

The paper is very well written, and generally very clear. In my opinion, the order of the sections is not logical and could be changed: the chemistry updates should be presented before the box model simulations and comparisons. The methodologies are appropriate, except for a minor reservation (see further below). I appreciate very much the detailed comparison of CMAQ with formaldehyde measurements and the discussion of discrepancies. I also appreciate the model implementation of bidirectional exchange of formaldehyde, justified on the basis of model comparisons with in situ hourly HCHO data. The overall results, including the cancer risks estimates, appear to be robust. Still, I think an acknowledgment of the remaining mechanistic uncertainties would be welcome in the implications sections, and possibly also in the Abstract.

We would like to thank the reviewer for the summary and comments. Responses to the specific comments are given below.

Main comments

1. I suggest to reorganize the sections in a more logical order. Currently, the results of the box model comparisons with the different mechanism are presented before discussing the mechanistic updates, which is weird. For example AMORE is mentioned in section 2 without telling what it is and how it relates to CRACMM2.

The box model results (originally Sect. 2; now Sect. 3) have been moved to follow the chemistry updates (originally Sect. 3; now Sect. 2) with some resulting minor edits in the text that are not detailed here.

2. CRACMM1 is evaluated against MCM v3.3.1, which is mostly fine. However, although MCM is indeed a very detailed mechanism, it has its limitations, which should be acknowledged. Isoprene oxidation experiments at Juelich have shown that the bulk rate of isomerisation of isoprene hydroxyperoxy radicals is underestimated in MCMv3.3.1, by as much as a factor of 3, with consequences for the production of HOx radicals but also, almost certainly, for HCHO production (since HCHO is produced along with MVK and MACR, which are overestimated in MCMv3.3.1) (Novelli et al., 2020). A crude but effective way to test this would be to modify the MCM mechanism following Novelli et al. (see their

Table 2). Note that there might be also caveats regarding the accuracy of MCM predictions for monoterpenes and for anthropogenic NMVOCs.

We have added a general caveat where MCM is first introduced in the text:

"While MCM is much more detailed than the chemical mechanisms typically used in CTMs, it has limitations and uncertainties."

We emphasize this with the following sentence already present in the manuscript:

"The box model simulations serve as a screening level identification of precursor systems that may not produce sufficient secondary HCHO in CRACMM1."

which we have reworded for clarity to the following:

"Deviations between MCM and CRACMM in the box model simulations serve as a screening process to identify precursors systems for further investigation and update in CRACMM."

The CRACMM2/AMOREv1.2 representation of isoprene chemistry is based on Wennberg et al. 2018 with additional updates. We have made the following updates in discussing the limitations of the isoprene chemistry in MCM:

"MCM has been previously found to underestimate the rate of isomerization of isoprene hydroxy peroxy radicals based on comparisons to experimental results (Novelli et al., 2020) which may also affect isoprene products, including HCHO. In addition to the comparisons to MCM, the production of HCHO from isoprene in CRACMM2 has also been compared to the more detailed representation from Wennberg et al. (2018) and compares favorably (see SI and Fig. S3). This is expected since CRACMM2 uses the AMOREv1.2 condensation of the Wennberg et al. (2018) isoprene mechanism (see Sect. 2.1)."

3. The estimate of cancer risks (and especially the controllable part) is likely underestimated, given the underestimations of HCHO abundances e.g. in oil and gas production areas. This should be better acknowledged in the implications section of the paper.

We have added the following to the implications section:

"Although the estimate of the number of cancer cases has increased with CRACMM2, the number may be underestimated since comparison to observations indicates that CRACMM2 is biased low. CRACMM2 was particularly biased low compared to TROPOMI HCHO in some western US oil and gas producing areas, including the Permian, Ft. Worth, and Anadarko Basins. CRACMM2 was also biased low in parts of California, including in the Los Angeles area (based on comparisons to TROPOMI) and in the Central Valley (based on comparisons to FIREX-AQ aircraft observations) which are two highly populated parts of the state."

Minor comments

l. 73 The last paragraph of the introduction should better present the structure of the paper.

We find that the last paragraph of the introduction follows the structure of the paper after the rearrangement of the chemistry updates and box modeling sections as suggested. We have made some

edits to this paragraph to specifically mention the HCHO deposition updates which were not previously mentioned here (new text in red). The manuscript structure with corresponding excerpts from the last paragraph of the introduction are given below:

Sect 2: We make several updates to CRACMM version 1 (CRACMM1), leading to CRACMM version 2 (CRACMM2). Most of the updates in CRACMM2 target HCHO, and additional updates for completeness are documented here for users of CMAQ and CRACMM2.

Sect. 3: Chemistry updates were screened with a box model, the Framework for 0-D Atmospheric Modeling (F0AM) (Wolfe et al., 2016a),

Sect. 4: and then tested in a series of regional CMAQ simulations covering the contiguous US (CONUS).

Sect. 5: The performance of CRACMM (1 and 2) in CMAQ is evaluated with a suite of observations including

Sect. 5.1: satellite based HCHO from TROPOspheric Monitoring Instrument (TROPOMI),

Sect. 5.2: observations from an aircraft campaign,

Sect. 5.3: and hourly surface observations from several field deployments.

Sect. 5.4: Based on the evaluation, sensitivity simulations are conducted to explore areas for future improvement of HCHO in CMAQ CRACMM. These include simulations with updates to HCHO dry deposition to reduce high nocturnal biases.

Sect. 6: Estimates of cancer risk from ambient exposure to HCHO derived from CMAQ CRACMM are provided along with an estimate of the portion of cancer risk that is controllable through reductions in anthropogenic $NO_X$ and ROC emissions.

l. 271 "autoxidation of PINAL and LIMAL": I suppose you mean PINALP and LIMALP, since pinonaldehyde and limonaldehyde do not undergo autoxidation.

Yes. This has been corrected.

l. 404-405: Decreases in HOx due to heterogeneous chemistry would indeed increase the lifetime of HCHO but also increase HCHO production from many VOCs including methane.

We have updated the explanation for the HCHO effects of $HO_2$ uptake to the following (new text in red):

"The updated heterogeneous chemistry results in small (<40 ppt) increases in HCHO in the southeastern US. There are two likely contributing factors. One is decreased $HO_X$ from uptake of $HO_2$ marginally increasing the lifetime of HCHO. The other is a decrease in the favorability of the $RO_2+HO_2$ channel with reduced $HO_2$ and resulting increase in the favorability of the $RO_2+NO$ channel which has higher HCHO yields compared to the $RO_2+HO_2$ route."

l. 440-442: The TROPOMI HCHO VCDs are scaled up by 25% when the column exceeds 8E15 molecules cm-2 to account for the low bias at high HCHO levels (De Smedt 2021). How does that compare with the

TROPOMI HCHO evaluation against FTIR data from e.g. Oomen et al. 2024 and Vigouroux et al. 2020 ? For very large columns, the discrepancy between FTIR and TROPOMI is likely much larger than 25%. The potential consequences for the comparisons presented here should be briefly discussed.

We have added the following:

> "The uniform scaling of 25% is a simplification. Previous comparisons of TROPOMI HCHO against ground-based Fourier-transform infrared (FTIR) observations of HCHO have found that the negative bias of TROPOMI at higher levels of HCHO increases with increasing FTIR HCHO (Vigouroux et al., 2020; Oomen et al., 2024). So, for areas with the highest HCHO, the correction of 25% scaling may still be too low."

l. 597 "Biogenic isoprene emissions could be low in CMAQ": I agree. How does the annual or seasonal emissions of isoprene from CMAQ compare with determinations based on MEGAN (see e.g. Kaiser et al. 2018) or from inverse modelling of satellite data (e.g. Muller et al., 2024) ?

We have added the following discussion of isoprene emissions:

> "The June–August 2019 total of isoprene emissions over the southeastern US (75-100 °W, 26-42 °N following Müller et al. (2024)) from the CMAQ inline implementations of BEIS (5.7 Tg C) and MEGAN (5.6 Tg C) are in good agreement. However, compared to an inversion optimizing isoprene emissions based on HCHO column totals from Ozone Monitoring Instrument (OMI) by Müller et al. (2024), the June–August total of isoprene emissions from BEIS over the southeastern US are slightly high (+8%) compared to an inversion where OMI HCHO was not bias corrected (5.3 Tg C) but low (-41%) compared to an inversion where OMI HCHO was corrected for a low bias in the OMI HCHO retrievals (9.7 Tg C). The emission totals reported here from the inverse analysis by Müller et al. (2024) are in a different year (2013) than our simulations (2019). While there is interannual variability in the emissions of isoprene, this would not account for such a large difference (-4.0 Tg C)."

Note that the emissions from Muller et al. (2024) were obtained from Figure 11c.

l. 629 and Fig. 6: I don't understand the separation of "bidirectional flux" and "leaf wetness deposition". Isn't deposition included in the representation of bidirectional flux? This should be clarified. Does the model of Shutter et al. account for the re-volatilization of formaldehyde when the leaf water evaporates (and when the Henry's law constant of HCHO decreases due to warming temperature), i.e. during daytime?

We have added the following additional descriptions of the bidirectional flux implementation based on Shutter et al. 2024 and the leaf-wetness based deposition:

> "We performed a sensitivity simulation for summer 2019 where the STAGE dry deposition model in CMAQ was updated to add a bidirectional flux for HCHO based on the HCHO stomatal compensation point parameterization of Shutter et al. (2024) and a relative humidity (RH) dependence to leaf wetness for dry deposition (Altimir et al., 2006; Burkhardt et al., 2009). The stomatal compensation point is taken as the internal concentration of HCHO in the leaf and represents the ambient HCHO concentration at which there is no net flux via the stomata. When the ambient concentration exceeds the compensation point, there is deposition. When the ambient concentration is below the compensation point, there is emission. The stomatal compensation point is then incorporated into

bidirectional flux calculations within the STAGE deposition module (see Clifton et al. (2023) for more details on the implementation of STAGE in CMAQ). The addition of the stomatal bidirectional flux parameterization of Shutter et al. (2024) tends to slightly increase HCHO (typically 0-50 ppt daily average). In addition to the stomatal bidirectional flux implementation, an additional sink was added by accounting for the role of plant surface wetness in deposition based on Altimir et al. (2006). Plant surface wetness was parameterized using RH based on experimental results by Burkhardt et al. (2009). This new surface wetness dependent deposition process decreased HCHO at night when RH is higher."

Technical comments

l. 80 "are evaluated" --> "is evaluated"

Yes. This has been corrected.

References

Kaiser, J., et al., High-resolution inversion of OMI formaldehyde columns to quantify isoprene emission on ecosystem-relevant scales: application to the southeast US, Atmos. Chem. Phys., 18, https://doi.org/10.5194/acp-18-5483-2018, 2018.

Muller, J.-F., et al., Bias correction of OMI HCHO columns based on FTIR and aircraft measurements and impact on top-down emission estimates, Atmos. Chem. Phys., 24, 2207, https://doi.org/10.5194/acp-24-2207-2024, 2024.

Novelli, A., et al., Importance of isomerization reactions for OH radical regeneration from the photo-oxidation of isoprene investigated in the atmospheric simulation chamber SAPHIR, Atmos. Chem. Phys., 20, 3333-3355, https://doi.org/10.5194/acp-20-3333-2020, 2020.

Oomen, G.-M., et al., Weekly-derived top-down volatile-organic-compound fluxes over Europe from TROPOMI HCHO data from 2018 to 2021, Atmos. Chem. Phys., 24, 449-474, https://doi.org/10.5194/acp-24-449-2024, 2024.

Vigouroux, C., et al., TROPOMI/S5P formaldehyde validation using an extensive network of ground-based FTIR stations, Atmos. Meas. Tech., 13, 3751, https://doi.org/10.5194/amt-13-3751-2020, 2020.

**Reviewer 2 (Narendra Ojha):**

The mechanism of secondary formaldehyde (HCHO) production has been updated in the regional air quality model – CMAQ. In addition, a night-time sink has been implemented. These updates helped model in better capturing the magnitudes in satellite and aircraft-based observations, and a diurnal change seen in field measurements.

Model results have been used to assess the health impact of HCHO on contiguous US. Manuscript presents significant advancements in chemistry and deposition of HCHO and is recommended for publication in ACP. However, following comments are offered for authors to consider during the revision.

We would like to thank the reviewer for the summary and comments. Responses to the specific comments are given below.

1. L.50-60: Add 1-2 statements on the uncertainty in anthropogenic and biogenic emissions of HCHO and precursors over the study region, from literature or by analyzing different emission inventories.

Uncertainty in emissions of HCHO and its precursors have been examined in a limited scope. See the response to comment 3 below for a comparison of biogenic precursor emissions. We did conduct a sensitivity analysis in which direct emissions of HCHO (from biogenic and anthropogenic sources) were doubled for a simulation covering June-August 2019. This resulted in localized increases but had little effect on the regional biases seen in the comparisons to TROPOMI HCHO. A scaling factor of 5x for direct HCHO emissions was required to reduce the mean bias against daily average HCHO observations from the Air Quality System (AQS) to near zero (see SI just before Fig. S14 for more about the AQS observations), but this 5x scaling was found to produce incorrect diurnal variability as compared to in situ measurements, and large negative biases against TROPOMI HCHO were still seen in the western US even with 5x primary HCHO scaling.

2. L.96-100: Add more rationale behind setting NOx with 1 ppb NO2. What were the initial mixing ratios of major HCHO precursors in box model? Wouldn't it be better to run 2-3 different cases representing urban and clean environments and then select all reactions to optimize chemistry for "regional-scale" modelling?

To the question about the rationale behind setting NO$_2$ to 1 ppb, we have made the following edits (new text in red):

"Simulations were run for 8 hours of photochemical processing with NO$_X$ initialized at an atmospherically relevant value of 1 ppb of NO$_2$ and allowed to evolve freely during the simulation."

"The box model setup employed here is limited in its ability to assess some atmospheric processes, such as transport or interactions between emissions from different sectors and does not capture the range of NO$_X$ and ROC precursor concentrations in the atmosphere."

To the question about the initial mixing ratios of major HCHO precursors, this can be gathered based on information in the current version of the manuscript. We have also added a sentence to direct readers to a figure in the SI which details the emissions values used to set the initial ROC precursor concentrations in the box model simulations. See below an excerpt from the manuscript with the new text shown in red:

"The concentrations of ROC precursors were initialized based on the emissions of each species with 100 Gg of annual emissions represented by 1 ppb (except for primary HCHO which was excluded). For the emission sector simulations (Fig. 1a), all emitted ROC species from each of 20 emissions sectors (Table S2) were initialized at their emission-weighted value. For the precursor system simulations (Fig. 1b), the total emissions across all sectors were divided into 19 distinct precursor groups (Table S3), and a simulation was conducted with initial concentrations for only the species belonging to a particular precursor group. Secondary HCHO from biogenic emissions was similarly assessed except that initial precursor concentrations were set with 1000 Gg of annual emissions represented as 1 ppb. Fig. S2 shows the annual emissions for each emission sector and precursor group which were used to set the initial ROC precursor concentrations."

With the description in the text and Fig. S2 (divide anthropogenic emissions by 100 Gg/ppb and divide biogenic emissions by 1000 Gg/ppb), readers can see how the initial ROC precursor concentrations were set for the emission sector and precursor group box model simulations.

To the question of running multiple cases spanning cleaner and more polluted conditions, we are not using the box model simulations here to optimize or tune the chemistry. As stated in the original version of the manuscript:

"The box model simulations serve as a screening level identification of precursor systems that may not produce sufficient secondary HCHO in CRACMM1."

which we have reworded for clarity to the following:

"Deviations between MCM and CRACMM in the box model simulations serve as a screening process to identify precursors systems for further investigation and updates in CRACMM."

The reviewer's suggestion is a reasonable approach, though we believe that type of approach may be better suited to a manuscript focused primarily on box modeling simulations. The current manuscript, however, is primarily focused on the chemical transport model, and that is where the bulk of the evaluation efforts were concentrated.

3. L.360: "Biogenic emissions……using the BEIS module". Since the biogenic emissions are major source for HCHO, some additional details should have been provided on this module / method. Difference with other widely used models like MEGAN and estimation on how much total biogenic emission it estimates (e.g. for isoprene) could be valuable.

We do not provide an overview of the BEIS module here. Further description of BEIS, including comparisons to MEGAN, are available in the Bash et al., 2016 reference given in the text:

Bash, J. O., Baker, K. R., and Beaver, M. R.: Evaluation of improved land use and canopy representation in BEIS v3.61 with biogenic VOC measurements in California, Geosci. Model Dev., 9, 2191-2207, 10.5194/gmd-9-2191-2016, 2016.

We do now provide a comparison of isoprene emissions from BEIS and MEGAN as implemented in CMAQ and a comparison to isoprene emissions derived from an inverse modeling analysis that used MEGAN for the prior emissions estimates. This is provided in Sect. 5.3 rather than in Sect. 4 which this comment references. The following was added:

"The June–August 2019 total of isoprene emissions over the southeastern US (75-100 °W, 26-42 °N following Müller et al. (2024)) from the CMAQ inline implementations of BEIS (5.7 Tg C) and MEGAN (5.6 Tg C) are in good agreement. However, compared to an inversion optimizing isoprene emissions based on HCHO column totals from Ozone Monitoring Instrument (OMI) by Müller et al. (2024), the June–August total of isoprene emissions from BEIS over the southeastern US are slightly high (+8%) compared to an inversion where OMI HCHO was not bias corrected (5.3 Tg C) but low (-41%) compared to an inversion where OMI HCHO was corrected for a low bias in the OMI HCHO retrievals (9.7 Tg C). The emission totals reported here from the inverse analysis by Müller et al. (2024) are in a different year (2013) than our simulations (2019). While there is interannual variability in the emissions of isoprene, this would not account for such a large difference (-4.0 Tg C)."

Note that the emissions from Muller et al. (2024) were obtained from Figure 11c.

4. L.382-385: Discuss how the chemical updates affected the mixing ratios of isoprene and monoterpene.

We have added the following:

"Isoprene itself is decreased in CRACMM2 compared to CRACMM1 because of increased reactivity. The summer average reductions in isoprene in the southeastern US are around 0.5 to 1 ppb (20-30%)."

"The two CRACMM monoterpene species are decreased due to slightly increased reactivity. In the southeastern US, reductions in CRACMM species API (which represents α-pinene, β-pinene, and other monoterpenes with one double bond) is reduced by around 50-100 ppt (5-15%) while reductions in CRACMM species LIM (which represents limonene and other monoterpenes with more than one double bond) is reduced by around 5-20 ppt (5-15%)."

5. Organization of the manuscript could be improved. The observational datasets and technical details (section 5.1) could be discussed earlier, before entering the results and discussions.

While the Introduction-Methods-Results and Discussion-Conclusions format is the more traditional approach, we found that organizing the methods and results by topic in Sect. 5 (TROPOMI satellite-based observations, FIREX-AQ aircraft observations, hourly surface observations, deposition updates) provided a better structure and improved readability for this manuscript since each of the datasets is

distinct and requires different methods for the analysis. As such, we have elected to retain the current structure of Sect. 5.

6. L.455-464: This is important result. The effect of emission uncertainty should be discussed in introduction section (as I have mentioned in my comment # 1). Large underestimation over California suggest that emissions / primary production may have large bias (and could have affected estimation of health effect). More discussion on this may be added.

There is some discussion of the potential role of emissions underestimation on the biases seen over California in the paragraph following the paragraph referenced in the comment:

"A large underestimate is seen in Arizona over the Tonto National Forest to the northeast of Phoenix. Large underestimates in California occur over the Los Angeles metropolitan area and over national forest land east of the Central Valley. Underestimates over the national forest land in Arizona and California could result from underestimated biogenic emissions. The underestimated HCHO in Los Angeles is more likely related to anthropogenic precursors and could result from either underestimated precursor emissions or secondary production. More detailed data and analysis of these individual areas, such as might be possible with a field campaign, are likely needed to explore the specific reasons for the underestimates of HCHO."

Additionally, based on this comment and a comment from the other reviewer, the following has been added to the Implications section (Sect. 6) to discuss the role of underestimated HCHO on the health risk analysis:

"Although the estimated number of cancer cases has increased with CRACMM2, the number may be underestimated since comparison to observations indicates that CRACMM2 is biased low. CRACMM2 was particularly biased low compared to TROPOMI HCHO in some western US oil and gas producing areas, including the Permian, Ft. Worth, and Anadarko Basins. CRACMM2 was also biased low in parts of California, including in the Los Angeles area (based on comparisons to TROPOMI) and in the Central Valley (based on comparisons to FIREX-AQ aircraft observations) which are two highly populated parts of the state."

7. Figure 5 – rightmost panel: check and make uniform the use of "southeastern" or "eastern US" between text and figures.

We have changed the figure panel title "eastern US" to "southeastern US" and added the following to the figure caption:

"Flights east of -97 °W were exclusively in the southeastern US and so are labeled as southeastern US in the rightmost panel title."